# Efficacy of the Treatment of Plantar Warts Using 1064 nm Laser and Cooling

**DOI:** 10.3390/ijerph19020801

**Published:** 2022-01-12

**Authors:** Elena de Planell-Mas, Blanca Martínez-Garriga, Miguel Viñas, Antonio J. Zalacain-Vicuña

**Affiliations:** 1Department of Clinical Sciences, School of Podiatry, Faculty of Medicine and Health Sciences, University of Barcelona, L’Hospitalet de Llobregat, 08907 Barcelona, Spain; elenaplanell@ub.edu; 2Department of Pathology & Experimental therapeutics, Faculty of Medicine and Health Sciences, University of Barcelona, IDIBELL, L’Hospitalet de Llobregat, 08907 Barcelona, Spain; blancamg13@gmail.com (B.M.-G.); mvinyas@ub.edu (M.V.)

**Keywords:** plantar warts, laser treatment, novel technique, cooling, viral genotype

## Abstract

Cutaneous plantar warts may be treated using several optional methods, with the use of laser surgery having increased in the last few years. This work examined the efficacy of laser treatment combined with simple cooling to reduce pain. The cure rate was approximately 84%. There were no significant differences in the efficacy of treatment for different viral genotypes. The laser parameters were 500 msec pulses, 30 W of power, and a fluence of 212 J/cm^2^ delivered in up to four sessions. Successful treatment was achieved after an average of 3.6 sessions.

## 1. Introduction

Cutaneous warts are non-malignant, intra-epidermal tumors that include common warts (verrucae vulgaris), plantar and palmar warts (verrucae plantares et palmares), flat warts (verrucae planae), and others [1]. Among them, plantar and palmar warts have the highest prevalence [2]. Currently, warts are diagnosed based on clinical findings. While they occasionally disappear spontaneously or remain asymptomatic for long periods of time [3,4], in most cases, treatment is sought.

Warts are most commonly caused by the human papilloma virus (HPV) [5], with cutaneous warts typically caused by HPV-2, -3, -7, -10, -27, and -57 (genus α), HPV-4, -60, and -65 (genus γ), and HPV-1 and -63 (genus µ) [2,6]. Of these, HPV-2, -27, -57, and HPV-1 are associated with plantar warts [6,7,8].

The treatment of plantar warts is a challenge for podiatrists. Current therapeutic strategies include chemotherapy (salicylic acid, glutaraldehyde, silver nitrate) and cytostatics (fluoracyl or bleomycin sulfate), but also cryotherapy, photodynamic therapy, and surgery [9,10,11]. Moreover, laser therapy has also been proposed and has become an indispensable therapeutic modality since its introduction in dermatology, along with the development of selective photothermolysis (use of light energy to selectively treat elements of the skin) [12].

Four general groups of visible-wavelength lasers are used in dermatology: pulsed lasers in the green and yellow millisecond domain, used to treat vascular malformations, hemangiomas, scars, photo-aging, proliferative lesions, and epidermal pigmented lesions; pulsed lasers in the red and near-infrared millisecond domain, used for the treatment of hypertrichosis and pigmented and deep venous lesions; low-power continuous green or yellow lasers, used for the treatment of superficial telangiectasias and for the photo-coagulation of lesions; and Q-switch lasers, used to remove tattoos and treat dermal melanocytosis, drug hyperpigmentation, and many pigmented lesions [13].

The photothermal effect is the best known. The wavelength determines the penetration into the tissues based on the absorption and scattering of light. Temperature increases when the power density increases, thus causing different tissue effects. As the temperature rises, the effects may begin with a transient hyperthermia, and subsequently, desiccation, protein denaturation and coagulation, tissue coagulation fusion, then tissue vaporization, and finally, its charring [14].

The 1064 nm laser seems to be suitable for wart treatment, thanks to the generation of a hyperthermic environment that has been shown to be effective in fighting HPV [15]. Histological studies have shown the coagulation and destruction of blood vessels in the papillary dermis in the wart region after laser irradiation [11].

Here, we report the clinical effectiveness of a procedure based on the use of laser at a wavelength of 1064 nm, in combination with cooling in order to reduce pain. We also consider the relation between the genotype of the etiologic virus and the effectiveness of laser treatment.

## 2. Materials and Methods

### 2.1. Design of the Work and Ethical Aspects

This was a prospective, observational, open study carried out at our center and without a control group. The main purpose was to investigate the efficacy of the 1064 nm laser in the treatment of plantar warts.

The study was performed in accordance with the recommendations of the Declaration of Helsinki. The protocol was previously approved by the Bioethics Committee of the University of Barcelona (IRB00003099). Informed consent was obtained from all patients who participated in the study.

### 2.2. Patients

Thirty-two patients clinically diagnosed with plantar warts were recruited in the laser unit of the Hospital de Podología de la Universidad de Barcelona, Spain. The clinical signs considered were the presence of divergent skin lines, small dark lesions, and hyperkeratosis. Wart tissue does not retain dermatoglyphics, and the application of compression laterally (pinch) produces pain [16]. The morphological characteristics of the papilloma lesions were carefully observed by dermatoscopy [17]. Inclusion criteria were male and female patients who had been clinically diagnosed with plantar warts not treated topically in the previous 3 months. Patients with neuropathy and vascular compromise were excluded. The recorded data included sex and age—with the patients classified into three groups (4–11 years, 12–20 years, and >21 years); the location of the warts on the right or left foot, heel (internal or external aspect), midfoot (internal or external aspect), forefoot (first, second, third, fourth, or fifth radius), and toes (first, second, third, fourth, or fifth); the time elapsed between the moment when the patient first became aware of the wart’s presence and when medical assistance was sought (≤6 months, 6–12 months, >12 months, or unknown); and the number of warts/patient.

### 2.3. Laser Instrumentation

The laser instrument was a 1064 nm laserS30 PODYLAS^TM^ (INTERmedic), Barcelona (Spain), and the parameters were as follows: pulse 500 msec, pause 1.0 sec, potency 30 W, and fluence 212 J/cm^2^. The minimal effective energy was 120 J, and the exposure time was 4 sec. These parameters were established after experimental work had been performed on cadaver specimens to evaluate the effect of vaporization produced by the laser on the area treated. The test method consisted of mimicking the therapeutic irradiation of a spot that was 4.0 mm in diameter, located in regions where warts usually appear (metatarsal and heel). A 1 cm^3^ sample of the vaporized region was taken, cut longitudinally, and the depth of vaporization was measured, with a depth between 6 and 8 mm consistently determined [18,19].

### 2.4. Cutaneous Warts Treatment

The treatment protocol is summarized in a flow diagram shown in Figure 1. Initially (t0), patients diagnosed with plantar warts completed a questionnaire, and their data were registered. The region surrounding the warts was disinfected using 1% chlorhexidine, and the hyperkeratinic surface was removed using a scalpel.

The wart was painted with India ink for targeted laser absorption [20]. Before the wart was irradiated, the area was cooled to reduce pain. Cooling was achieved by placing a frozen non-colored gel (*Gel UltrasonidoTransparente*, Versus Medico Kft Budapest Hungary) on a glass slide, which was then covered with a coverslip and frozen using an ethyl chloride spray. The frozen slide was placed on the wart prior to its irradiation (Figure 2).

In addition to the anti-pain effect, the use of the slide improves the uniformity of the wart surface. The laser beam is directed at an angle of 90°, which allows for an optimal, uniform focal distance. The hand piece of the laser had a spot diameter of 4 mm. For a wart surface >4 mm in diameter, the area was divided to cover the full surface (Figure 3). After laser treatment, the region was carefully observed using a dermatoscope. In the case of persistent papillae, irradiation was repeated until the subcutaneous tissue was visible. If a serous exudate developed due to vaporization, antiseptic solution (povidone iodine) and a protective bandage were applied. Finally, if the plantar wart was in a pressure area, a felted offloading device was applied to prevent overload and support.

Patients were instructed to wash and dry the lesion and to apply antiseptic (povidone iodine) daily using a dressing in case an exudate developed.

A follow-up examination was performed one week after treatment to clinically evaluate the lesion (Figure 4), remove surface hyperkeratosis, and monitor the eventual reappearance of dermatoglyphics. Treatment was considered to be successfully finalized when papillae were no longer visible under dermatoscopy. If small ulcers were observed, treatment aimed at tissue regeneration was prescribed, and the patients would be re-examined weekly until a complete cure was achieved. For those with remaining papillae, a new round of laser treatment was administered. A maximum of five laser treatment sessions were performed. If the warts persisted, alternative treatment was prescribed for the sixth visit.

### 2.5. Viral Genotypes

The HPV genotypes of the treated warts were also determined. The tissue obtained after hyperkeratosis removal was used for DNA extraction following a previously described phenol-chloroform-isoamyl alcohol method (Green and Sambrook, 2012). The concentration and purity of the DNA were determined spectrophotometrically (NanoDrop Spectrophotometer ND-1000). Thereafter, the DNA was analyzed using the SK-polymerase chain reaction (SKPCR) method introduced by Sasagawa and Mitsuishi (2012), which detects most of the common HPV types causing cutaneous warts. After DNA sequencing, the HPV types were identified using the BLAST server (https://blast.ncbi.nlm.nih.gov/blast (accessed on 12 August 2019) [8].

### 2.6. Statistics

A descriptive statistical analysis was conducted for all of the variables. Continuous variables were reported according to the number of valid cases and the mean and standard deviation (SD). Categorical variables were described according to the absolute and relative frequencies of each category compared to the total number of relevant values (N). For absent values, their number was described by group. Comparisons were carried out using an ANOVA and a bilateral statistical significance level of 0.05. All analyses were carried out on the data set, using all available information, in accordance with the intention to treat criteria. The statistical analyses were performed using the SAS program, version 9.2.3.

## 3. Results

This section may be divided into subheadings. It should provide a concise and precise description of the experimental results and their interpretation.

The 32 patients included in the study had an average age of 36.19 (±17.77) years, and 65.6% were female. A single wart was treated in 84.4% of the patients, two warts in 9.4%, and three or more warts in 6.2%, for a total of 41 warts. The genotypes identified in the 41 samples were HPV-1 (14.63%), HPV-2 (19.51%), HPV-27 (17.07%), HPV-57 (41.46%), and HPV-65 (7.32%).

The effectiveness of the laser treatment of plantar warts is reported in Table 1, which shows the healing rate according to the viral genotype. Among the different genotypes, the cure rate differed and was the lowest for patients with warts caused by the HPV-2 genotype (*p* = 0.0315). The differences in the rates of change over the course of the treatment were small.

Cure was achieved in 85.5% of the laser-treated plantar warts (Table 2). There were differences in the healing of warts at different locations; up to 100% cure for warts in the heel area, followed by those located on the toes (92.3%), midfoot (75%), and forefoot (69.2%).

An analysis of the warts according to their diameter and response to treatment is shown in Table 3. Cure was achieved in 83.3% of the warts with a diameter ≤4 mm. For larger warts, the cure rate was quite similar at 87%.

## 4. Discussion

The cooling method used in conjunction with laser enabled the use of a focal length appropriate for the targeted spot on the skin surface. The optimal laser parameters were: 500 msec pulse, 30 W of power, and a fluence of 212 J/cm^2^. With these settings, pulse emission resulted in the delivery of an energy of 15 J to the tissue. With an emission energy of 120 J, treatment was effective without causing intolerable pain to the patient, and an average of 3.6 sessions per wart were needed.

Kimura et al. [11] and Han et al. [21] reported the results obtained using a 1064 nm laser for the treatment of skin warts, including plantar warts. In both studies, a handpiece with a spot diameter of 5 mm was used. In the Kimura study, the application parameters comprised a pulse duration of 15 msec and a fluence of 150–185 J/cm^2^. Six sessions were administered, with an interval of 4 weeks between sessions. The overall cure rate was 56%, and that of plantar warts specifically was at 39% [11]. In the study of Han et al. [21], the pulse duration was 20 msec, and the fluence was 200 J/cm^2^. Treatment was administered in four sessions, with an interval of 4 weeks between sessions. The cure rate was 96% [21]. In both studies, pain was avoided by applying ice before treatment and topical anesthetic ointment or 1% lidocaine infiltration. Bacelieri et al. [22] used a 585 nm pulsed dye laser treatment and obtained global cure values of 48–93% for warts located in different skin areas. In the study of Tan et al., using the same type of laser, the cure rate was 50% in patients with plantar warts [23]. Neither the diameter of the warts nor their zone of appearance seemed to influence healing. According to the literature, spontaneous remission occurs in 4–8% of cases in a period of six months [24]; thus, we expect 2 or 3 spontaneous cures in the group studied. Our results pointed out slight differences in the rate of healing, depending on the location of the warts. A higher proportion of unsuccessful treatment (30%) was seen in the forefoot and midfoot (25%), while in areas submitted to higher pressures such as the heel or toes, the treatment failed at a lower proportion (0% and 7%, respectively).

Plantar warts may be caused by different HPV genotypes, as we pointed out in the case of the patients of our service [8]. Warts of the genotype HPV-1a of the genus µ, HPV-65 of the genus γ, and HPV-27 and -57 of the genus α were the most effectively treated using the 1064 nm laser. However, for genotype HPV-2, also of the α-genus, cure could not be demonstrated, either because the patients abandoned treatment or surgery was necessary due to unfavorable healing. In our patients, plantar warts caused by the HPV-2 genotype evolved over >6 months. Whether warts differing in their genotype respond differently to different laser parameters has not been determined and was not considered in our study since the number of cases were not enough to obtain conclusions. Another parameter considered was the size of the warts. In this regard, it seems that rates of healing do not depend upon the size of the wart (Table 3). Assuming that bigger warts should also be older, this result suggests that the time of evolution is probably not a critical condition.

## 5. Conclusions

Our study showed that plantar warts can be treated using a 1064 nm laser for an average of three to four treatment sessions, and that any resulting pain can be successfully treated using a cooling method. The treatment parameters consisted of 500 ms pulses, 30 W power, and 212J/cm^2^ fluence. This protocol was particularly effective for the treatment of the plantar wart genotypes: HPV-1a, -27, -57, and -65. Therefore, it is possible that the efficacy of the cure may not depend only on the treatment used but also on the wart genotype, which should be considered.

## Figures and Tables

**Figure 1 ijerph-19-00801-f001:**
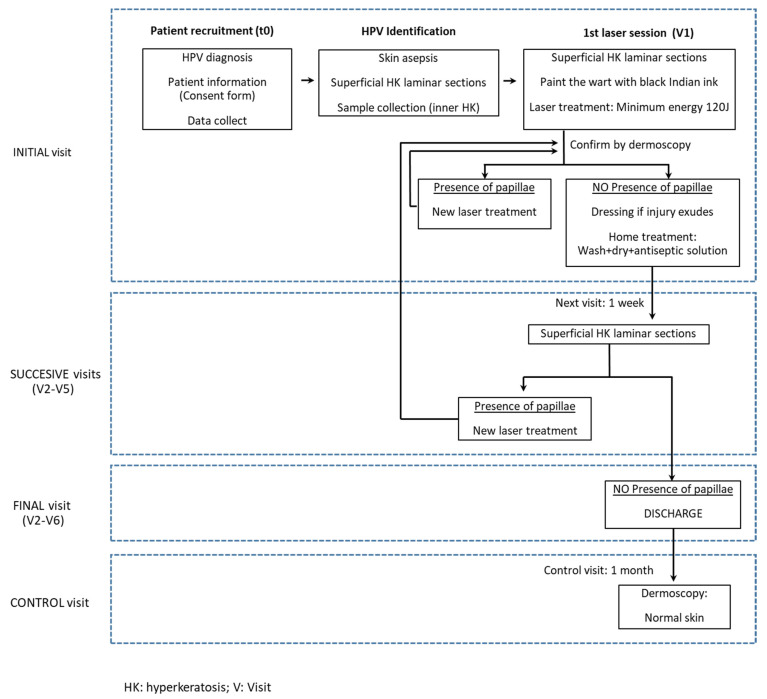
Flux diagram for the diagnosis and laser treatment of plantar warts.

**Figure 2 ijerph-19-00801-f002:**
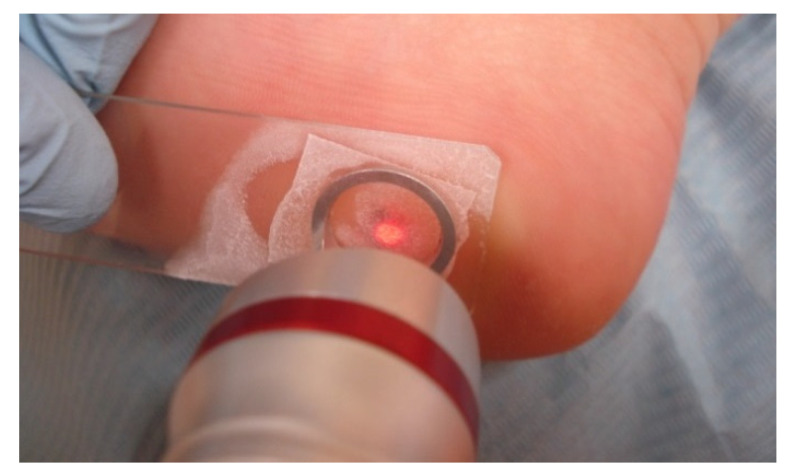
Wart treatment using a heel-targeted laser and in situ cooling.

**Figure 3 ijerph-19-00801-f003:**
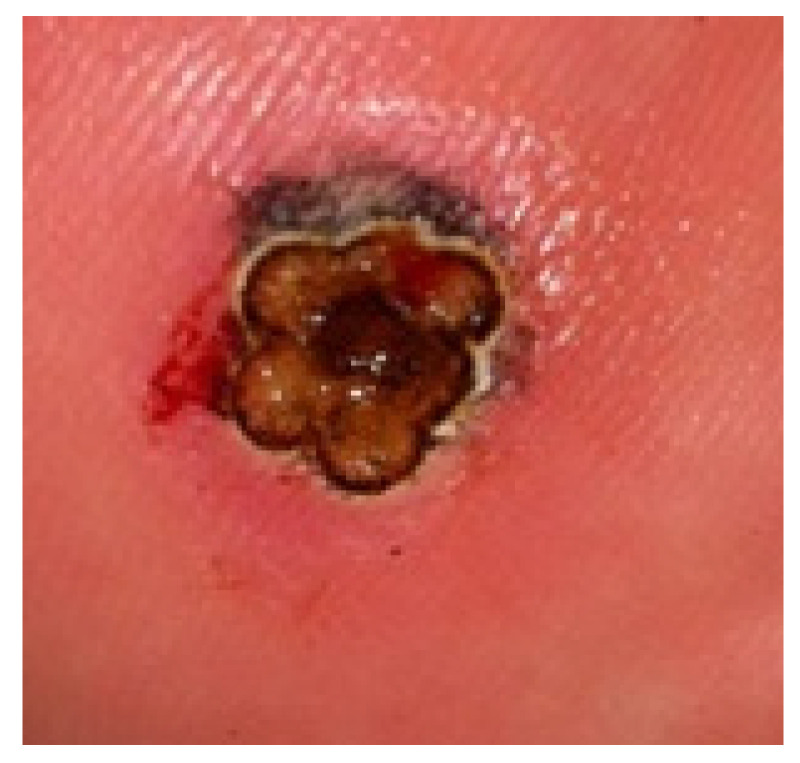
Plantar wart with a diameter >4 mm. Simulation of the distribution of the laser application zones ensuring coverage of the whole lesion.

**Figure 4 ijerph-19-00801-f004:**
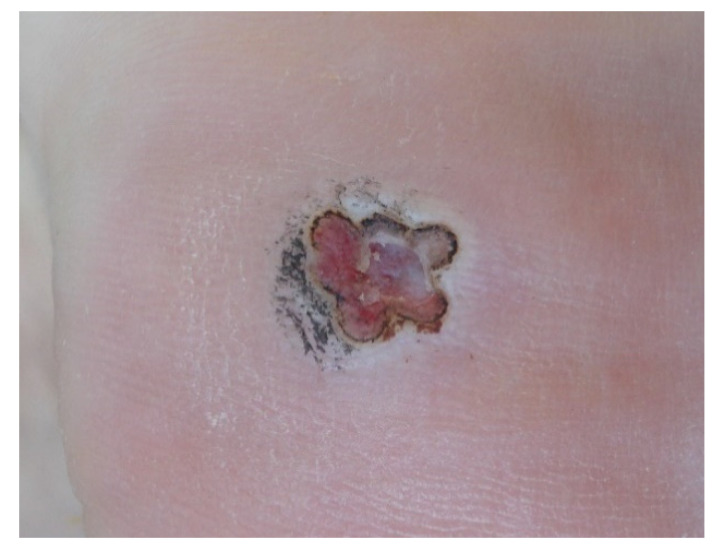
A plantar wart >4 mm in diameter one week after treatment.

**Table 1 ijerph-19-00801-t001:** Warts healed by laser treatment.

Variable		Total (*n* = 41)	HPV-1 (*n* = 6)	HPV-2 (*n* = 8)	HPV-27 (*n* = 7)	HPV-57 (*n* = 17)	HPV-65 (*n* = 3)	*p*-Value (1)
Warts	Yes	41	6	8	7	17	3	
	Total	41	6	8	7	17	3	
Healed warts	Yes	35	6	4	6	16	3	
	No	6		4	1	1		
	Total	41	6	8	7	17	3	0.0315
Healing achieved at:	Visit 2	7	1	2	1	3		
	Visit 3	10	1	1	2	5	1	
	Visit 4	12	2		3	5	2	
	Visit 5	1				1		
	Visit 6	5	2	1		2		
	Total	35	6	4	6	16	3	0.8653
Change of treatment	Yes	4		3		1		
	No	37	6	5	7	16	3	
	Total	41	6	8	7	17	3	0.0608
Treatment abandonment	Yes	2		1	1			
	No	39	6	7	6	17	3	
	Total	41	6	8	7	17	3	0.4525

**Table 2 ijerph-19-00801-t002:** Healed warts according to their location.

Variable		Total (*n* = 41)	Forefoot (*n* = 13)	Midfoot (*n* = 4)	Toes (*n* = 13)	Heel (*n* = 11)	*p*-Value (1)
Warts	Total	41	13	4	13	11	
	Yes	41	13	4	13	11	
Healed warts	Yes	35	9	3	12	11	
	No	6	4	1	1		
	Total	41	13	4	13	11	0.1422
Healing achieved at:	Visit 2	7	2	2	2	1	
	Visit 3	10	1		3	6	
	Visit 4	12	5	1	4	2	
	Visit 5	1			1		
	Visit 6	5	1		2	2	
	Total	35	9	3	12	11	0.3636
Change of treatment	Yes	4	2	1	1		
	No	37	11	3	12	11	
	Total	41	13	4	13	11	0.4275
Treatment abandonment	Yes	2	2				
	No	39	11	4	13	11	
	Total	41	13	4	13	11	0.2098

**Table 3 ijerph-19-00801-t003:** Healed warts according to their diameter.

Variable		Total	≤4 mm	>4 mm	*p*-Value (1)
Warts	Yes				
	Total	41	18	23	
Healing warts	Yes	35	15	20	
	No	6	3	3	
	Total	41	18	23	0.7446
Healing achieved at:	Visit 2	7	4	3	
	Visit 3	10	4	6	
	Visit 4	12	6	6	
	Visit 5	1	1		
	Visit 6	5		5	
	Total	35	15	20	0.2029
Change of treatment	Yes	4	2	2	
	No	37	16	21	
	Total	41	18	23	0.7959
Treatment abandonment	Yes	2	1	1	
	No	39	17	22	
	Total	41	18	23	0.8586

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
