# Peer review of "Efficacy of the Treatment of Plantar Warts Using 1064 nm Laser and Cooling"

_ijerph, 2022, doi:10.3390/ijerph19020801_

Round 1

Reviewer 1 Report

In their work “Efficacy of the treatment of plantar warts using 1064-nm laser and cooling” de Planell-Mas et al. applied a systematic approach to prospectively  demonstrate the influence of a laser treatment on plantar warts. The procedure is described in a precise way and the manuscript should be considered for the publication in IJERPH. Nevertheless, a minor revision is recommended based on the following items.

  1. Line 34: The authors could furthermore cite the curing mechanism behind applying NIR-lasers for plantar warts. For instance the advantages of using 1064 nm lasers especially for the treatment of plantar warts are low scattering effects/deep penetration depth as well as thermal effects.
  2. Line 64: It should be mentioned that this system is based on a Nd:YAG laser source.
  3. Line 83 : In Figure 1 “Sample collaction” and “Desmatoscopy” should be corrected.
  4. Line 126 : Was a pre-analysis of the data regarding normal distribution conducted? If there was no normal distribution, an equivalent method like K-W-ANOVA should be applied.
  5. Line 132: The sentence seems to be discontinued. Please check again.
  6. Line 142: Please change VPH to HPV in Table 1.
  7. Line 159: The number of average treatments is presented to be 2.7. Based on the informations in Table 1 and 2 the weighted arithmetic mean of the number of treatments would be: 3.6. Please reconsider.
  8. Line 154-181 (Discussion): Since there is no non-treated control group, the authors should cite from the literature if usually no spontaneous remission is expected for the maximum time of this study.

Author Response

  1. Line 34: The authors could furthermore cite the curing mechanism behind applying NIR-lasers for plantar warts. For instance the advantages of using 1064 nm lasers especially for the treatment of plantar warts are low scattering effects/deep penetration depth as well as thermal effects.

The introduction has been modified on the basis of the reviewer’s comments: the following paragraphs have been added:

  1. “Moreover, laser therapy has also been proposed and has become an indispensable therapeutic modality since its introduction in dermatology along with the development of selective photothermolysis (use of light energy to selectively treat elements of the skin)”
  2. “Four general groups of visible wavelength lasers are used in dermatology: Pulsed lasers in the green and yellow millisecond domain, used to treat vascular malformations, hemangiomas, scars, photo-aging, proliferative lesions, and epidermal pigmented lesions, pulsed lasers in the red and near infrared millisecond domain, used for the treatment of hypertrichosis and pigmented and deep venous lesions, low power continuous green or yellow lasers, used for the treatment of superficial telangectasias and for the photo-coagulation of lesions,  and Q-switch lasers used to remove tattoos and treat dermal melanocytosis, drug hyperpigmentation, and many pigmented lesions.”
  3. Photothermal effect is the best known. The wavelength determines penetration into the tissues based on the absorption and scattering of light. Temperature increases when the power density increases causing different tissue effects. As the temperature rises, the effects may begin with a transient hyperthermia and, subsequently desiccation, protein denaturation and coagulation, tissue coagulation fusion, then tissue vaporization and finally its charring

  1. Line 64: It should be mentioned that this system is based on a Nd:YAG laser source.

The recommendation has been followed by adding  the sentence :   “The 1064nm laser, seems to be  suitable for wart treatment thanks to the generation of a hyperthermic environment that has been shown to be effective in fighting HPV [Smith]. Histological studies showed coagulation and destruction of blood vessels in the papillary dermis in the wart region after laser irradiation”

  1. Line 83: In Figure 1 “Sample collaction” and “Desmatoscopy” should be corrected.

Spelling has been revised along the whole ms. We appologize for the mistakes

  1. Line 126: Was a pre-analysis of the data regarding normal distribution conducted? If there was no normal distribution, an equivalent method like K-W-ANOVA should be applied.

The normality of data was not determined by any pre-analysis tests. Given the sample size of the study and the fact that the objectives were descriptive and purely exploratory, no specific adjustment or test was  considered to be needed in this regard. This is an exploratory study lacking previous hypothesis, with this n, non-parametric tests are not needed.

  1. Line 132: The sentence seems to be discontinued. Please check again.

This sentence has been removed following the comment of reviewer 2

  1. Line 142: Please change VPH to HPV in Table 1.

We apologize for the mistake. It has been changed.

  1. Line 159: The number of average treatments is presented to be 2.7. Based on the informations in Table 1 and 2 the weighted arithmetic mean of the number of treatments would be: 3.6. Please reconsider.

We sincerely apologize. The reviewer is right and the correct value is 3.6, it has been changed in all sections of the manuscript.

  1. Line 154-181 (Discussion): Since there is no non-treated control group, the authors should cite from the literature if usually no spontaneous remission is expected for the maximum time of this study.

According to the literature spontaneous remission is in between 4 and 8 % of cases in periods of six months.  This has been included in the main text.

Reviewer 2 Report

In this manuscript, de Planell-Mas et al. describes a trial in which a set of patients was recruited who manifested HPV-induced plantar warts. The authors describe the outcomes from using laser therapy to treat these warts.

The report is fairly clear and interesting, but the authors do a very poor job of placing their study in the context of what else is known about laser therapy for warts. A reader would have a lot of questions that are not addressed by the manuscript.

Specific points below:

Line 34 “However, laser therapy has also been proposed.” Authors should give much more detail here. Have lasers been used before for treating benign skin tumors? Malignant skin tumors? Other defects like actinic keratosis? Hypertrophic scar? Better explanation of the state of laser use in dermatology is critical for readers to put this study in context. Authors should expand the introduction to include some of this information to better contextualize their study.

Line 56-57 what was the rationale for these age groupings? Were these decided on prior to patient enrollment?

Line 64-66 Just want to confirm, the authors say the exposure time was 4 seconds. Does this mean that the authors performed 8 pulses with 7 1-second pauses between?

Line 131-132 should be removed from the template

Line 145-146 How does the claim of statistically significant differences in the healing of warts in different locations square with the lack of significant P values in table 2?

Authors perform statistical analyses to show that the healing rate of warts in different locations, resultant from different HSV genotypes, and in different sizes do or do not differ. The authors should discuss all of this in the discussion, as well as propose potential reasons why there may have been differences, and place this in the context in what has been published in the literature.

Author Response

  1. Line 34 “However, laser therapy has also been proposed.” Authors should give much more detail here. Have lasers been used before for treating benign skin tumors? Malignant skin tumors? Other defects like actinic keratosis? Hypertrophic scar? Better explanation of the state of laser use in dermatology is critical for readers to put this study in context. Authors should expand the introduction to include some of this information to better contextualize their study.

It has been done, details are given in the answers to reviewer 1.

  1. Line 56-57 what was the rationale for these age groupings? Were these decided on prior to patient enrollment?

The age groups were made once the patients were recruited. The 3 age groups were based on the life stages by age: infancy / childhood, adolescence and adulthood. It has been also indicated in the main text.

  1. Line 64-66 Just want to confirm, the authors say the exposure time was 4 seconds. Does this mean that the authors performed 8 pulses with 7 1-second pauses between?

Yes, this is exactly as we did.

  1. Line 131-132 should be removed from the template

Done

  1. Line 145-146 How does the claim of statistically significant differences in the healing of warts in different locations square with the lack of significant P values in table 2?

The p-values ​​shown on the paper are completely exploratory in nature and we completely  agree with the reviewer  as how they should be interpreted. Although the main results are very strong and conclusive, as long as they are in agreement with previously published results, there is no possible hypothesis test to test.

Authors perform statistical analyses to show that the healing rate of warts in different locations, resultant from different HSV genotypes, and in different sizes do or do not differ. The authors should discuss all of this in the discussion, as well as propose potential reasons why there may have been differences, and place this in the context in what has been published in the literature.

Several sentences have been added to the discussion following this recommendation.  

Round 2

Reviewer 2 Report

Authors have addressed my concerns adequately